# Total Intravenous Anaesthesia with Ketamine, Medetomidine and Midazolam as Part of a Balanced Anaesthesia Technique in Horses Undergoing Castration

**DOI:** 10.3390/vetsci8080142

**Published:** 2021-07-26

**Authors:** Alexandra Cunneen, Shaun Pratt, Nigel Perkins, Margaret McEwen, Geoffrey Truchetti, Joanne Rainger, Trish Farry, Lisa Kidd, Wendy Goodwin

**Affiliations:** School of Veterinary Science, The University of Queensland, Gatton, QLD 4343, Australia; shaun.pratt@outlook.com.au (S.P.); n.perkins1@uq.edu.au (N.P.); mcbent@gmail.com (M.M.); gtruchetti@gmail.com (G.T.); j.rainger@uq.edu.au (J.R.); t.farry@uq.edu.au (T.F.); l.kidd@uq.edu.au (L.K.); w.goodwin@uq.edu.au (W.G.)

**Keywords:** horse, balanced anaesthesia, midazolam, castration

## Abstract

To evaluate the use of ketamine-medetomidine-midazolam total intravenous infusion as part of a balanced anaesthetic technique for surgical castration in horses. Five healthy Standardbred cross colts were premedicated with IV acepromazine (0.01–0.02 mg/kg), medetomidine (7 µg/kg) and methadone (0.1 mg/kg) and anaesthesia induced with IV ketamine (2.2 mg/kg) and midazolam (0.06 mg/kg). Horses were anaesthetised for 40 min with an IV infusion of ketamine (3 mg/kg/h), medetomidine (5 µg/kg/h) and midazolam (0.1 mg/kg/h) while routine surgical castration was performed. Cardiorespiratory variables, arterial blood gases, and anaesthetic depth were assessed at 5 to 10 min intervals. Post-anaesthesia recovery times were recorded, and the quality of the recovery period was assessed. The anaesthetic period and surgical conditions were acceptable with good muscle relaxation and no additional anaesthetic required. The median (range) time from cessation of the infusion to endotracheal tube extubation, head lift and sternal recumbency were 17.2 (7–35) min, 25 (18.9–53) min and 28.1 (23–54) min, respectively. The quality of anaesthetic recovery was good, with horses standing 31.9 (28–61) min after the infusion was ceased. During anaesthesia, physiological variables, presented as a range of median values for each time point were: heart rate 37–44 beats/min, mean arterial pressure 107–119 mmHg, respiratory rate 6–13 breaths/min, arterial partial pressure of oxygen 88–126 mmHg, arterial partial pressure of carbon dioxide 52–57 mmHg and pH 7.36–7.39. In conclusion, the co-administration of midazolam, ketamine and medetomidine as in IV infusion, when used as part of a balanced anaesthetic technique, was suitable for short term anaesthesia in horses undergoing castration.

## 1. Introduction

Equine veterinarians frequently surgically castrate colts and stallions, with most surgeries performed under field conditions using general anaesthesia [1,2,3]. For equine veterinarians to maintain surgical efficiency and enhance personnel and patient safety, it is important the anaesthetic technique chosen for field castrations be safe, convenient and without the need for cumbersome equipment, easy to perform and provide adequate analgesia and muscle relaxation. Total intravenous anaesthesia (TIVA) is commonly chosen for this purpose as the duration of anaesthesia required for this procedure is short (<60 min), minimal equipment is required and numerous studies have reported various combinations of IV infusions of commonly ketamine, an alpha-2 agonist (e.g., xylazine, medetomidine or detomidine) and a muscle relaxant (e.g., guaifenesin, or benzodiazepines such as diazepam and midazolam) [4,5,6,7]. In addition to castrations, TIVA is also commonly used in equine practice for other short surgical or diagnostic procedures performed in the field and hospital.

Recently, the commercial formulation of guaifenesin, a centrally acting muscle relaxant, was unavailable in countries such as Australia, the United States and Canada and continues to have limited availability worldwide [5,8]. Consequently, there has been considerable interest in finding an alternative to guaifenesin in order to provide adequate skeletal muscle relaxation for TIVA with co-infusions of ketamine and an alpha-2 agonist.

This research group recently compared an infusion of ketamine, medetomidine and midazolam with an infusion of ketamine, medetomidine and guaifenesin to maintain anaesthesia for 50 min in 14 young horses undergoing computerised tomography (CT) in a randomised, blinded, cross over study [9]. Both infusions produced a clinically comparable quality of anaesthesia, however, recovery from anaesthesia was of a better quality following the infusion of ketamine, medetomidine and midazolam [9]. While both infusions appeared equipotent, a major limitation of this study was that horses were not subjected to a nociceptive stimulus, as would be common in clinical practice, and there are currently no reported studies investigating this infusion for surgical procedures in horses at these infusion rates.

In order to be clinically relevant, we chose to assess the suitability of the ketamine, medetomidine and midazolam, administered at the same infusion rates used in the aforementioned study, as part of a balanced anaesthetic technique for routine castration in horses. Balanced anaesthesia, as described by John Lundy in 1926 [10], refers to a combination of anaesthetic agents and techniques (premedication, regional anaesthesia, general anaesthesia) to achieve the desired objectives of anaesthesia, which include; analgesia, muscle relaxation, amnesia and reduction or elimination of autonomic reflexes while maintaining homeostasis. This balanced anaesthetic approach also enables a reduction in individual anaesthetic agent doses, minimising their associated undesirable side effects [11]. In addition to the anaesthetic technique previously reported [9], to provide clinical applicability for equine veterinarians, this study included additional premedication and co-induction drugs and intra-testicular local anaesthetic, as commonly used in the field when performing such castration procedures. We hypothesised that an infusion of ketamine, medetomidine and midazolam, when used as part of a balanced anaesthetic technique at the current infusion rates, would be suitable to maintain anaesthesia in colts for routine surgical castration.

## 2. Materials and Methods

### 2.1. Horses

A total of 5 Standardbred cross colts; weighing (median {range}) 324 (319–390) kg and aged 14.2 (14.1–14.5) months were scheduled for elective surgical castration. All colts were owned by the institution and were required to undergo surgical castration as per the institution’s animal welfare and husbandry management protocols regardless of any associated research or investigations. Four colts had previously been anaesthetised twice for a non-invasive CT study and 1 colt had not been anaesthetised. A minimum period of at least 1 month had passed since horses had been previously anaesthetised. All procedures were performed by approval of the University of Queensland Production and Companion Animals Ethics Committee (AEC number 286/10). Horses were considered healthy on the day of anaesthesia based on the results of a physical examination.

### 2.2. Anaesthesia

The skin overlying the left jugular vein was clipped and aseptically prepared and, following the subcutaneous deposition of 20 mg of 2% lidocaine (Lignomav, Mavlab Pty Ltd., Slacks Creek, QLD, Australia), a 14G 3 inch catheter (BD Angiocath^®^, Becton Dickinson Rowa Australia, Virginia, QLD, Australia) was placed for IV administration of anaesthetic drugs. Horses were premedicated IV with 0.01–0.02 mg/kg acepromazine (A.C.P 10, Ceva Animal Health Pty Ltd., Gelnorie, NSW, Australia), 7 µg/kg medetomidine (Domitor^®^, Zoetis Australia Pty Ltd., Rhodes, NSW, Australia) and 0.1 mg/kg methadone (ilium methadone, Troy Laboratories Pty Ltd., Glendenning, NSW, Australia). Five minutes later, anaesthesia was induced with 2.2 mg/kg ketamine (Ketamil^®^, Troy Laboratories Pty Ltd., Glendenning, NSW, Australia) and 0.06 mg/kg midazolam (Hypnovel^®^, Roche Products Pty. Ltd., Sydney, NSW, Australia) IV. Anaesthesia was then maintained for 40 min with an IV infusion of ketamine 3 mg/kg/h, medetomidine 5 μg/kg/h and midazolam 0.1 mg/kg/h in 0.9% NaCl at a rate of 2 mL/kg/h delivered via an infusion pump (Plum A+ Infusion System, Hospira Inc., Lake Forest, IL, USA). To make the infusion, 13.75 mL was withdrawn from a 500 mL 0.9% NaCl bag and 7.5 mL ketamine (100 mg/mL), 1.25 mL medetomdine (1 mg/mL) and 5 mL midazolam (5 mg/mL) was added to the bag. Once recumbent, endotracheal (ET) intubation with a cuffed 22 mm internal diameter ET tube was performed to maintain patent airways. The ET tube remained uncuffed during anaesthesia to prevent further reducing tracheal lumen diameter with spontaneous ventilation. If horses became apnoeic (no breath for 60 s), this cuff was to be inflated to allow positive pressure ventilation. The horse was supplied with oxygen at 15 L/min by insufflation with a catheter positioned in the cranial third of the ET tube. Horses were positioned in dorsal recumbency on the floor of the induction box for surgery. Following aseptic preparation of the surgical site, 10 mL of 2% lidocaine was injected into the parenchyma of each testicle using a 21G 1.5 inch needle. At completion of surgery, horses were positioned in right lateral recumbency until the infusions were ceased. Additional anaesthetic was administered if horses were too lightly anaesthetised as evidenced by gross purposeful movement of the head or limbs as previously described [9].

### 2.3. Anaesthesia Monitoring

Prior to premedication, baseline measurements of heart rate (HR) and respiratory rate (fR) were recorded. Following induction of anaesthesia, a 20G 1.4 inch (Optiva^®^, Smiths Medical International Ltd., Rossendale, Lancashire, UK) over-the-needle catheter was placed in the transverse facial artery for blood pressure monitoring and blood gases sampling. The urinary bladder was aseptically catheterised to permit continuous drainage throughout anaesthesia and to minimise urinary bladder distension during recumbency.

Cardiorespiratory variables and clinical signs of anaesthetic depth were initially assessed 10 min following induction of anaesthesia and then every 5 min throughout anaesthesia. A multiparameter monitor (Datex AS3, Datex-Ohmedia, General Electric Healthcare, Helsinki, Finland) measured nasopharyngeal temperature, arterial haemoglobin saturation (SpO_2_) and a continuous base apex lead II electrocardiogram (ECG) trace monitored HR and rhythm. Direct arterial blood pressure was measured with the transducer positioned at the level of the sternal manubrium and zeroed to atmospheric pressure. A new transducer was used for each horse. The respiratory rate was determined by counting thoracic wall excursions over 1 minute. Arterial blood samples were collected anaerobically at 10 min intervals following induction of anaesthesia. Arterial samples were immediately analysed using a blood gas analyser (Gem Premier 3500 with iQM^®^, Instrumentation Laboratory, Lexington, MA, USA) and temperature corrected to record pH, PaO_2_, PaCO_2_, HCO_3_^−^, lactate and haematocrit. A demand valve connected to an oxygen cylinder was available if horses became apnoeic (no breath for 60 s) or arterial blood gas analysis showed PaO_2_ < 60 mmHg or PaCO_2_ > 65 mmHg. All monitoring equipment used for each horse was regularly calibrated and serviced as per manufacturing guidelines, based on the international quality assurance standards.

### 2.4. Surgery

Surgery commenced approximately 3 minutes after administration of intra-testicular lidocaine. The castration surgeries were performed by an experienced veterinary surgeon or by veterinary students under the direct supervision of a surgeon using an open castration technique as previously described [3]. The total surgery time (time of first incision to end of surgery) was recorded. Post-operative analgesia was provided with phenylbutazone, initially 4.4 mg/kg IV (Nabudone P, Troy Laboratories Pty Ltd., Glendenning, NSW, Australia) after recovery from anaesthesia, and then 2.2 mg/kg SID PO (Bute Paste, Randlab Pty Ltd., Chipping Norton, NSW, Australia) for 3 days.

### 2.5. Recovery from Anaesthesia

The time from induction of anaesthesia to the start of the infusion and then the time to the end of infusion were recorded. Additionally, the time taken from the end of the infusion to when horses were extubated (first swallow), first lifted their head, achieved sternal recumbency and stood were recorded. During recovery, oxygen insufflation was continued via a catheter positioned in the nasopharynx until it was dislodged due to movement of the horse.

All anaesthesia recoveries were recorded onto a digital video recording device and retrospectively scored on 2 separate occasions (28 days apart) by a panel of 3 specialist veterinary anaesthetists. Video clips presented to the evaluators were edited to begin at endotracheal extubation and end approximately 5 min after each horse successfully stood. The evaluators were blinded to the treatment and each horse was de-identified by black and white filtering of the video recording. The order that the anaesthesia recoveries were presented to evaluators was randomized on both occasions using the Excel randomization function (Microsoft^®^ Excel Version 16.12, Microsoft, Redmond, WA, USA).

Three anaesthesia recovery scoring systems were used (see Appendix A, Appendix B and Appendix C). Due to its widespread use in the literature, in combination with the recommendations by Suthers et al. [12], a composite scoring system (CSS), modified by the authors from that originally described by Donaldson [13], was validated for use as a grading scale in this study. This CSS devalued subcategories of the lateral and sternal phases whilst heavily weighting those associated with the standing phase, as the latter is considered highly correlated with soft tissue and orthopaedic injuries during recovery in horses. This culminated into a total score calculation between 13 and 122.5 to quantify the recovery quality, with a higher score indicating a poorer recovery quality (please refer to Appendix A). The second recovery scoring system employed was a simple descriptive scale (SDS), modified from that originally described by Young and Taylor [14] and further modified by Vettorato [15] and Ray-Miller [16]. The overall quality of recovery was assessed on a scale of 1-excellent to 5-unacceptable (please refer to Appendix B). Thirdly, a visual analogue scale (VAS), adapted from Hubbell [17], entailed placing a “cross” on the continuous 10 cm line scale to represent the overall recovery score from a 0-perilous to 100-perfect recovery (please refer to Appendix C).

### 2.6. Statistical Analysis

Physiological variables (HR, fR, SpO_2_, temperature, blood pressure, blood gases, pH and lactate), anaesthesia recovery times and anaesthesia recovery scores were non-normally distributed and reported as median and range. Physiological variables at the designated monitoring time points were compared with baseline values where available or at the 10 min time point using a Friedman test with Dunn’s multiple comparisons test (*p* < 0.05). Calculations were performed with GraphPad Prism (www.graphpad.com; version 9 [accessed on 11 January 2018]) and Microsoft Excel (Version 16.51, Microsoft^®^ Corporation, Brisbane, QLD Australia [accessed on 15 July 2021]).

## 3. Results

### 3.1. Anaesthesia and Surgery

One colt was fractious and difficult to handle and received 200 mg xylazine 90 min prior to premedication to facilitate catheter placement. Due to investigator miscommunication, another colt did not receive medetomidine (7 µg/kg IV) until 5 min after the acepromazine and methadone injections. In all horses, induction of anaesthesia was unremarkable, and the median (range) time for horses to become laterally recumbent was 1.0 (0.8–1.3) min and the infusion of ketamine, medetomidine and midazolam began 3.3 (1.5–6.0) min after induction of anaesthesia.

Surgery commenced 12 (10–14) min after induction of anaesthesia and lasted 12 (10–16) min. All horses demonstrated spontaneous blinking and lacrimation intermittently during anaesthesia; however, this was not associated with gross purposeful movement. Surgical conditions were subjectively assessed by the surgeons to be adequate in all cases and no comments regarding hind leg or cremaster muscle tension were made. No additional anaesthetic was administered to any horse. The duration of anaesthesia (induction of anaesthesia to cessation of the infusion) was 41.5 (40–45) min.

### 3.2. Physiological Data

Cardiorespiratory variables are shown in Table 1. Mean arterial blood pressure for all horses measured 90 mmHg or above across all anaesthesia durations. There were no episodes of apnoea, and controlled pressure ventilation was not required. Arterial blood gas variables are shown in Table 2. PaO_2_ was significantly increased at 40 min when compared with 10 min (*p* = 0.041).

### 3.3. Anaesthesia Recovery

The time from cessation of the infusion to extubation, head lift, sternal recumbency and standing was 17.2 (7–35) min, 25 (18.9–53) min, 28.1 (23–54) min and 31.9 (28–61) min, respectively. Anaesthesia recovery scores are reported in Table 3. The CSS overall score of 30.5 (24.8–40.5) represents a good to favourable recovery as it falls within the lower 20% of the score range of 13 to 122.5 (13 representing an optimal recovery). The SDS score of 2 (2–2.5) represents a ‘good smooth recovery’ with a score of 1 representing an ‘excellent smooth recovery’ and a score of 5 representing a ‘very poor to unacceptable recovery’. The VAS score of 85 (76.5–91) also represents a good to favourable recovery as it falls within the upper 20% of the score range, with a score of 100 representing a perfect recovery. (See Appendix A, Appendix B and Appendix C below.)

## 4. Discussion

This study demonstrated that ketamine, medetomidine and midazolam, when infused as part of a balanced anaesthetic technique, provided adequate anaesthesia conditions for surgical castration. This was evidenced by horses not moving during the procedure and the fact that no additional anaesthetic was required. Previously, the same ketamine, medetomidine and midazolam infusion technique administered at the same infusion rates as this study had been used to maintain anaesthesia for 50 min in horses undergoing a non-invasive procedure [9]. Interestingly, approximately 30% of horses moved and required additional anaesthetic despite no surgical or nociceptive stimulus being applied. These horses were premedicated with medetomidine and induced with ketamine intravenously prior to commencing on their randomly allocated TIVA protocols. Horses in this study received additional premedication to medetomidine with acepromazine and methadone, co-induction drugs of ketamine and midazolam and intra-testicular lidocaine, thus suggesting that a balanced anaesthetic technique improved the quality of anaesthesia.

Horses were premedicated with a combination of acepromazine, methadone and medetomidine. Acepromazine, although possessing no analgesic properties, has been shown to decrease the minimum alveolar concentration of inhalational anaesthetic agents [18]. Consequently, it is not unreasonable to assume that it may also reduce the requirements of intravenous drugs used for maintenance anaesthesia. Methadone, an opioid mu agonist, has been shown to have analgesic properties in horses [19] and has been included in standing anaesthesia protocols to reduce the dose and infusion rate of the co-infused alpha-2 agonist [20,21]. While no sedation scoring was undertaken in this study, it is likely that when these drugs were combined with the same dose of medetomidine used in the previous study, horses were more sedated prior to induction of anaesthesia [9]. Consequently, horses may have had reduced anaesthetic requirements and, therefore, were more deeply anaesthetised with the same infusion rate of ketamine, medetomidine and midazolam.

Additionally, midazolam was co-administered with ketamine for induction of anaesthesia. This was not conducted previously [9] as it was thought that it may confound comparison between the two treatment groups. Midazolam, however, does not appear have an effect on reducing the rates of additional anaesthetic administration when administered for co-induction with ketamine prior to commencement of an infusion of ketamine, alpha-2 agonist, and midazolam used in TIVA [5,6,7]. Nonetheless, midazolam administered with ketamine for co-induction of anaesthesia in this study may have aided in establishing higher plasma concentrations of midazolam prior to commencement of the infusion and contributed to the balanced anaesthetic technique.

In horses, all alpha-2 adrenergic agonists available for administration to horses have been used for balanced anaesthesia due to their sedation and potent analgesic effects [22]. Although not currently registered for horses in countries such as Europe, Australia and the United States, medetomidine, including its active enantiomer dexmedetomidine, have been shown to be short-acting, have high clearance rates and undergo rapid distribution, compared to other alpha-2 agonists such as xylazine, making them more suitable for intravenous infusions [23,24,25,26,27]. This is in addition to producing similar post-anaesthetic recovery times and recovery qualities in horses as compared to xylazine [24,27]. These pharmacokinetic properties of medetomidine led to its choice as both a premedication and co-infusion agent in this study.

Lidocaine used as a local anaesthetic adjunct to intravenous anaesthesia in horses undergoing castration has been shown to reduce the amount of incremental anaesthetic doses required to maintain anaesthesia compared with a saline control [28]. Additionally, local anaesthetic administration two minutes prior to castration in horses anaesthetised with isoflurane and medetomidine has been reported to improve the overall quality of the peri-anaesthetic period and reduce postoperative pain [29]. In this study, it is unlikely that the lidocaine provided a complete nociceptive blockade as it was not administered subcutaneously over the incision site, and previously it has been reported that intra-testicular lidocaine may not diffusely distribute to the cremaster muscle when locally infused into the testis [30]. This is also in consideration of the time allowed between the local infusion of lidocaine into each testis and the first skin incision being only 3 min compared to the standard 5 min. Regardless, it is likely that the multimodal analgesic approach used in this study, including the intra-testicular administration of lidocaine, reduced nociception during surgery, facilitated surgery by reducing muscle tone, and may have improved the quality recovery from anaesthesia.

As previously mentioned, Pratt et al. [9] reported improved recoveries in horses that had received an infusion of ketamine, medetomidine and midazolam compared with an infusion of ketamine, medetomidine and guaifenesin. In 14 horses, the quality of recovery from anaesthesia with ketamine, medetomidine and midazolam was described as “good smooth recovery to average, fair recovery” (score of 2.5) for the SDS and scores in the 65th and 60th percentile for optimal recovery for the CSC and VAS, respectively. In this study, the quality of recovery was slightly improved with recovery from anaesthesia classified as a “good smooth recovery” (score 2.1) for the SDS and scores in the 80th percentile for optimal recovery for the CSC and VAS. While the number of horses in this study is too small to make definitive conclusions regarding the improved quality of recovery from anaesthesia, the results are comparable to other studies that have reported a good quality of recovery from anaesthesia following infusions of ketamine, midazolam and an alpha-2 agonist [5,6,7].

Due to the lack of supportive equipment and facilities, the maintenance of adequate cardiopulmonary function is important for any anaesthetic technique that may be performed in the field. In these colts, HR was maintained within acceptable ranges for anaesthetised horses and MAP did not fall below 90 mmHg in any horse for the duration of the infusion [11]. Previously, studies investigating the cardiovascular effects of TIVA with ketamine, midazolam and medetomidine or xylazine have reported good haemodynamic stability during the infusions with HR, MAP and cardiac output not differing significantly from baseline values in horses anaesthetised for 60–70 min [5,6].

Pulmonary function during the infusion was similar to that reported by Pratt et al. [9]. During the infusion, all horses breathed spontaneously, although some respiratory depression was observed, as evidenced by mild hypercapnia (median PaCO_2_ 54–57 mmHg). Intratracheal oxygen insufflation at 15 L/min was sufficient to prevent hypoxaemia (PaO_2_ < 60 mmHg). Respiratory depression and hypoxaemia have been reported in horses receiving ketamine, alpha-2 and midazolam or guaifenesin-based TIVA infusion regimens in horses [6,31,32,33]. Certainly, in the absence of oxygen supplementation, infusions of ketamine, midazolam and medetomidine or xylazine have resulted in hypoxaemia [5,6]. These findings, in concert with the current study and Pratt et al. [9], highlight the importance of oxygen supplementation in horses, even for short procedures.

Finally, it has been suggested that the ideal anaesthetic technique for castration in horses should be cost-effective as well as providing adequate surgical conditions, analgesia, and a safe and predictable peri-anaesthetic period [28]. At the time of writing, the cost of midazolam in Australia compared with guaifenesin, to make 1 L of equipotent infusions4 is approximately $25 AUD and $123 AUD, respectively [34]. This represents a significant economic saving when using the midazolam infusion regimen. As countries such as the United Kingdom release licensed formulations of midazolam for equine anaesthesia [35], this is likely to facilitate improved availability and cost efficiency of this benzodiazepine for veterinarians.

A limitation of this study was that only a very small number of horses were enrolled and, therefore, the study may have been statistically underpowered. Unfortunately, this was a clinical study, and only a small number of horses were available in the designated study period. Additionally, the sample population was quite homogenous and, therefore, may not be representative of the general horse population. Furthermore, a more diverse range of surgeries to determine the suitability of the balanced anaesthetic technique for different nociceptive stimuli would be clinically beneficial.

## 5. Conclusions

In conclusion, this clinical trial demonstrated that the co-administration of midazolam, ketamine and medetomidine as in IV infusion, when used as part of a balanced anaesthetic technique, was suitable for short term anaesthesia in horses undergoing routine castration. This anaesthetic technique may provide a useful and cost-effective alternative to combinations that include guaifenesin. Further work with a larger number of horses undergoing various surgical procedures is required to more extensively assess the suitability of this anaesthetic regimen for surgery in the field.

## Figures and Tables

**Table 1 vetsci-08-00142-t001:** Cardiorespiratory variables for five colts anaesthetised for 40 min with infusions of ketamine, medetomidine and midazolam as part of a balanced anaesthetic technique for surgical castration.

Variable	Time (Minutes) after Induction of Anaesthesia							
	Baseline	10	15	20	25	30	35	40
HR (beats/min)	44 (40–80)	39 (32–39)	44 (34–60)	43 (34–60)	43 (35–59)	38 (34–43)	37 (30–44)	36 (30–43)
*f*_R_ (breaths/min)	20 (16–24)	6 (4–29)	7 (5–17)	13 (5–21)	12 (6–15)	10 (5–15)	11 (5–15)	11 (5–15)
SAP (mmHg)	-	140 (124–150)	145 (128–158)	136 (129–156)	134 (127–138)	132 (126–142)	131 (123–141)	131 (123–143)
DAP (mmHg)	-	104 (84–109)	104 (86–113)	105 (87–110)	96 (91–103)	98 (90–104)	93 (85–103)	95 (83–103)
MAP (mmHg)	-	116 (101–126)	118 (102–127)	119 (102–126)	112 (106–117)	110 (106–117)	110 (104–119)	107 (100–117)
SpO_2_ (%)	-	96 (95–97)	96 (95–98)	96 (95–98)	96 (95–98)	97 (85–97)	98 (97–98)	97 (97–98)

Heart rate (HR), respiratory rate (*f*_R_), systolic arterial pressure (SAP), mean arterial pressure (MAP), diastolic arterial pressure (DAP) and arterial haemoglobin saturation (SpO_2_). Results as median (range).

**Table 2 vetsci-08-00142-t002:** Arterial blood gas variables for four ^#^ colts anaesthetised for 40 min with infusions of ketamine, medetomidine and midazolam as part of a balanced anaesthetic technique for surgical castration.

Variable	Time (Minutes) after the Start of IV Infusion			
	10	20	30	40
pH	7.36 (7.33–7.38)	7.38 (7.35–7.39)	7.38 (7.35–7.40)	7.39 (7.35–7.40)
PaCO_2_ (mmHg)	54 (51–60)	52 (50–57)	54 (51–58)	57 (54–57)
PaO_2_ (mmHg)	88 (76–145)	97 (77–127)	122 (81–245)	126 ^a^ (117–261)
HCO_3_^−^ (mmol/L)	30.0 (28.5–33.9)	30.9 (28.2–33.5)	31.6 (29.5–35.1)	33.2 (31.5–34.7)
Lactate (mmol/L)	1.6 (1.5–1.9)	1.9 (1.8–2.0)	1.8 (1.8–2.5)	1.6 (1.5–1.8)

^a^ Statistically different from 10 min (*p* < 0.05) ^#^ Perioperative arterial blood gas analyses for one weanling were not undertaken. Arterial partial pressure of oxygen (PaO_2_), arterial partial pressure of carbon dioxide (PaCO_2_), bicarbonate (HCO_3_^−^). Results as median (range).

**Table 3 vetsci-08-00142-t003:** Anaesthetic recovery scores using a Composite Score System, Simple Descriptive Scale and Visual Analogue Scale for five colts anaesthetised for 40 min with infusions of ketamine, medetomidine and midazolam as part of a balanced anaesthetic technique for surgical castration.

	Median Score
Composite Score System	
A. activity in recumbency (score 1–5)	2 (1–2)
B. move to sternal (score 1–10)	1 (1–5)
C. attempts to sternal (score 1–10)	3 (1–7.5)
D. sternal phase (score 1–10)	3 (1–3)
E. move to stand (score 1–10)	3 (1–4.5)
F. strength (score 1–10)	3 (1–4.5)
G. attempts to stand (score 1–10)	2 (1–5)
H. balance and coordination (score 1–10)	2 (2–3)
I. knuckling (score 1–5)	1 (1–2)
J. overall attitude (score 1–10)	2 (1–5)
K. overall recovery (score 1–10)	3 (2–4)
Overall calculated score	30.5 (24.8–40.5)
**Simple descriptive scale** (score 1–5)	2 (2–2.5)
**Visual Analogue Scale** (0–100 mm)	85 (76.5–91)

**Composite score system**: Overall calculated score = 2 × (E + F + G + H) + I + J + K + 0.5 × (A + B + C + D). Minimum calculated score 13 (ideal recovery) and maximum calculated score 122.5 (worst recovery). **Simple descriptive Scale**: 1 (excellent, smooth recovery) to 5 (very poor, unacceptable recovery). **Visual Analogue Scale**: 0 mm perilous recovery to 100 mm perfect recovery. For full details of scoring systems see Appendix A, Appendix B and Appendix C. Results as median (range).

## Data Availability

Datasets analysed in this study are available upon request from the corresponding author. The data is stored on UQRDM and mediated access is available upon request.

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
