# Peer review of "Total Intravenous Anaesthesia with Ketamine, Medetomidine and Midazolam as Part of a Balanced Anaesthesia Technique in Horses Undergoing Castration"

_vetsci, 2021, doi:10.3390/vetsci8080142_

Round 1
Reviewer 1 Report
Comments to the Authors
Many thanks for the opportunity to review this interesting manuscript.
Major points:
The discussion regarding clinical relevance, cost effectiveness and practical aspects are welcome and I would suggest adding an additional segment discussing the choice of medetomidine as alpha 2 adrenoceptor agonist. Specifically why it was chosen instead of licensed products like xylazine and detomidine and the resulting legal considerations regarding use of unlicensed drugs.
Calibration of equipment: please provide information about calibration or servicing of equipment used. Especially regarding arterial pressure measurement and arterial blood gas analysis. Were new transducers used for arterial pressure measurement in every case?
Modification of recovery scoring system: as recovery quality is one of the main results of this study it would be beneficial to discuss the modifications and validation of the recovery scoring used in the main text of the manuscript and not in the appendix.
Minor points
Line 88 and 114: consider removing “100%” as this is inaccurate and stating the exact concentration of oxygen is not relevant.
Line 171-172: Consider removing “well maintained” from this sentence. Mean arterial pressure measured at 90 mmHg or above is a stronger statement on it’s own.
Line 190: remove # after “four” in the title of Table 2.
Line 226: please review – is “4” meant to be used as reference?
Line 269: please review for gramatical errors
Line 329: referencing inconsistent with main text of manuscript and the paper Suthers et al. 2011 is not listed as reference.
Lines 358, 365: as above
Reviewer 2 Report
Reviewer comments for manuscript ID vetsci-1254988 entitled ‘Ketamine, Medetomidine and Midazolam Intravenous Infusion as Part of a Balanced Anaesthesia Technique to Maintain Anaesthesia in Horses Undergoing Castration’
General comments
It is an important work on balanced anaesthesia in equines and is a useful contribution to the dynamic process of researching better anaesthetic protocols for equine anaesthesia. Manuscript is well written with minimum errors. Statistical treatment of data is robust except a very low number of horses used in the study. However, further studies in larger number of subjects can further strengthen the results of this study. The use of three scales for anaesthetic assessment further strengthen the use of the protocol in future studies /surgeries. I have few queries that I have mentioned in my comments. This manuscript is a useful contribution towards equine anaesthesia in field conditions.
Specific comments
Line 53: Balanced anaesthesia also means the reduction of the toxic side effects all the drugs used in the combination. Please add this to the definition.
Lines 206-12: Please explicitly clarify these sentences here as to how these two infusion techniques differed. Rewrite the protocols for readers’ comprehension.
Lines 225-233: This discussion shows that midazolam has non-significant role in the protocol. Please clarify and it can be suggested that further studies are needed to explore it further.
Lines 234-45: It is possible that Lidocaine masks the effect of other anaesthetics. The analgesia and reduced muscle tone attributed to the infusion of anaesthetics might be due to lidocaine induced analgesia. Please clarify.
Lines 267-76: Do you advocate, based on your study, this protocol to be administered for equine castration in field conditions where facilities for supplemental oxygen might not exist?
Reviewer 3 Report
Reviewer comments on Manuscript number: animals-1254988
The present manuscript shows a total intravenous anesthesia drug combination administered by constant rate infusion, in horses undergoing castration
The manuscript is not very well written, and the information showed lacks of novelty, however, the study adds some information that may be useful considering TIVA techniques for field anesthesia and surgery in horses. Some parts of the manuscript need major clarification and clarification before the opportunity to be published.
Broad comments
Title
L2-4. The title that better describes this study could be “Ketamine-Medetomidine-Midazolam constant rate infusion as the main components of total intravenous anesthesia in horses undergoing castration.
Introduction
L34. Please replace “anaesthetic” for “anaesthesia”
L28-L59. The introduction lacks of a background information related to the use of TIVA and its main components in horses. The question of investigation is not clear. It seems that, the limited availability of guaifenesin triggered the study, instead of looking for alternatives to inhaled anesthesia in horses, or looking for its suitability for surgery or for improving recovery when comparing to other drug combinations, or the way that premed drugs plus intratesticular lidocaine may helped to achieve a surgical level of anesthesia.
Material and methods
Even when this research has the approval of an ethic committee, this reviewer ask the authors if this project followed the CONSORT guidelines. If so, separately please attach the flow diagram.
L71. No blood work on these horses before anesthesia?
L84-86. What was the final volume and drug concentration after dilution with 0.9% NaCl ?
L87. Characteristics of the Endotracheal tube?
L88. What did you mean when referring to “insuflation”?
L90. What was the criteria for using 10 mL of lidocaíne to each testicle? Please specify the way and procedure of administration.
L92. How many horses needed additional doses of anesthesia? In addition, what was the final dose administered to them?
L106. What ECG lead was employed?
Results
L173. What was the criteria for not using mechanical ventilation? You showed ETCO2 values larger than 45 mm Hg and PO2 values lower than 80 mm Hg in some horses in table 2.
L187-195. Table 2 and 3. Statistical differences are missing in these tables. Please add.
Discussion and conclusion
As the question of investigation or the aim of the study Is not clear, the entire discussion section has to be rewritten.
Thank you.
Reviewer 4 Report
the aim of the paper is to describe the efficacy of a TIVA protocol with ketamine medetomidine and midazolam for castration in horses. the paper is overall well presented, I could find only some spelling mistakes here and there in the manuscript.
However I think that, despite the results, this can only be considered a preliminary study because of the extremely low number of animals and the numerous variables that were involved. They also lack of a control and comparison are only made with literature.

Round 2
Reviewer 1 Report
Many thanks for your consideration of all suggestions and the changes you have made to the manuscript.
Author Response
Thank you very much for your further review.
Reviewer 3 Report
Reviewer comments on Manuscript number: VetSci-1254988 R2
The manuscript has been improved however, some parts of the manuscript need clarification before the opportunity to be published.
Broad comments
Title
L2-4. The title that better describes this study could be “Ketamine-Medetomidine-Midazolam constant rate infusion as the main components of total intravenous anesthesia in horses undergoing castration.
Response 1: Thank you for your comment. We, the authors, feel it would be over-interpreted to state in the title that we are just investigating the TIVA protocol for horses undergoing castration. Midazolam and its clinical practicality in this protocol is not just as part of the IV infusion; there is also the induction midazolam that needs to be factored in. Furthermore, we feel describing a balanced anaesthetic technique is important, firstly as this was what had been performed for these horses as a multimodal balanced anaesthetic protocol. Secondly, other key components especially the intra-testicular lidocaine and premedication methadone for analgesia cannot be dismissed.
A revised title such as “Total Intravenous Anaesthesia with Ketamine, Medetomidine and Midazolam as a Part of a Balanced Anaesthesia Technique in Horses Undergoing Castration” may be more appropriate.
- This reviewer agrees with the authors, thanks for the clarification.
Introduction
L28-L59. The introduction lacks of a background information related to the use of TIVA and its main components in horses. The question of investigation is not clear. It seems that, the limited availability of guaifenesin triggered the study, instead of looking for alternatives to inhaled anesthesia in horses, or looking for its suitability for surgery or for improving recovery when comparing to other drug combinations, or the way that premed drugs plus intratesticular lidocaine may helped to achieve a surgical level of anesthesia.
Response 3: There is a component of assumed knowledge about equine field anaesthesia techniques and that TIVA is only ever used in such circumstances for practicality means. Inhalational anaesthesias in horses are logistically only performed in hospitals. This has been amended in the revised manuscript to further inform the reader. The main focus of the study was to investigate this midazolam based TIVA protocol, as part of a balanced anaesthetic technique, at these dose rates for a routine surgical procedure (ie castration) in horses, as would be performed by equine veterinarians worldwide. This research group had previously investigated this midazolam TIVA protocol as an alternative to a guaifenesin-based TIVA protocol. However, a key limitation was the lack of any noxious stimuli provided to the horses whilst under anaesthesia. In the field setting, horses are very rarely anaesthetised for procedures that do not cause noxious or painful stimuli. This study aimed to address this limitation at the dose rates used in the original investigation to provide an evaluation into its clinical applicability.
In terms of the other possible investigations for this research, controls would need to have been required to appropriately look into comparisons of recovery or the impacts of the premedication and local anaesthetic agents on anaesthetic quality. Therefore, the gap in knowledge for the wider research community and literature that this study addresses is the use of this balanced anaesthetic technique, including the intra-testicular lidocaine and premed methadone, utilised at these dose rates, for routine surgical procedures such as castrations.
- Thank you for your clarification
Material and methods
Even when this research has the approval of an ethic committee, this reviewer ask the authors if this project followed the CONSORT guidelines. If so, separately please attach the flow diagram.
Response 4: Please see the attached CONSORT flow diagram as requested. This study was not performed as a randomized controlled trial (RCT) as is the criteria for CONSORT checklists and reporting. This study is more appropriately categorized as a preliminary trial secondary to the original RCT published in Equine Veterinary Journal in 2019 by this research group. Therefore the second half of the flow diagram is not applicable as the intervention or anaesthetic protocol was not randomly allocated.
Furthermore, ethics approval was granted to investigate this anaesthetic technique and protocol. On the basis of the institution’s welfare and animal husbandry practices, these horses were to have undergone castration regardless of any research associated. This is why such a small sample size was utilised as the anaesthetic protocol investigation was not the priority for these horses undergoing castration at the time.
- Thank you for your clarification
L71. No blood work on these horses before anesthesia?
Response 5: No, as was the decision of the institution at the time. This could be discussed in the discussion as a limitation of the study. Please see the revised manuscript.
- Thank you for your clarification
L84-86. What was the final volume and drug concentration after dilution with 0.9% NaCl ?
Response 6: The final volume and drug concentration of ketamine, medetomidine and midazolam in the 1L NaCl bags is dependent upon the body weight (kg) of each horse. Every horse received a new, individually calculated 1L IV infusion bag based on their weight at the dose rates outlined in the study design (lines 84-87) to enable IV administration at 2ml/kg/hr. Please clarify how this information is required for the reader’s understanding.
- What this reviewer ask to the authors is to add enough information about the way you prepare your CRI drugs, as the TIVA administration of analgesic-anesthetic drugs is one of the goals of your study. Then you could mention for example.. “….by adding 5 mL of a 5 mg/mL midazolam to a 1 L of NaCl 0.9% and administered at a rate of 2 mL/kg/hr...” Please add the information needed to make this procedure easier for reader’s understanding and practical implementation.
L87. Characteristics of the Endotracheal tube?
Response 7: Each horse was intubated with a cuffed 22mm internal diameter endotracheal tube to maintain patent airways. They remained uncuffed throughout anaesthesia to prevent reducing tracheal airway diameter with spontaneous ventilation. However, if the
demand valve or mechanical ventilation were required if horses became apnoeic, this cuff would have been inflated to allow for positive pressure ventilation. This information can be included in the revised manuscript.
- Thank you.
L88. What did you mean when referring to “insuflation”?
Response 8: A small diameter oxygen line was positioned with the cranial third of the ET tube to supplement oxygen during anaesthesia. This information can be included in the revised manuscript.
-Thank you.
L90. What was the criteria for using 10 mL of lidocaíne to each testicle? Please specify the way and procedure of administration.
Response 9: A 21g 1.5inch needle inserted directly into the parenchyma of the testicle was performed to administer 10ml of 2% lidocaine for each testicle. This information can be included into the revised manuscript.
-Thank you.
L92. How many horses needed additional doses of anesthesia? In addition, what was the final dose administered to them?
Response 10: As stated in lines 167-168 in the results section, no additional anaesthetic was administered to any horse throughout anaesthesia in this study.
-Thank you.
L106. What ECG lead was employed?
Response 11: Base-apex ECG was utilised. This information can be included in the revised manuscript.
- It is important to add if Lead II or others were also monitored. Thank you.
Results
L173. What was the criteria for not using mechanical ventilation? You showed ETCO2 values larger than 45 mm Hg and PO2 values lower than 80 mm Hg in some horses in table 2.
Response 12: The aim was to adhere to field conditions for these horses undergoing castration where appropriate, with the overriding notion to intervene when the health of the horses would be compromised. Mechanical ventilation, with the exception of demand valves in some general practices, is logistically only practical in hospitals. This raises the very important question of how compromised are horses that undergo field anaesthetics with limited monitoring equipment and interventions available to the clinicians performing them? This is outside the scope of this study and manuscript. Yes, there were PaCO2 and PaO2 values approaching outside the acceptable ranges however, as discussed in lines 267-276, this highlights the importance of, at the very least, supplemental oxygen.
- This reviewer thanks for the above clarification, however, for readers and considering that this protocol is intended for field anesthesia, the cut-off limits for SpO2, PaCo2 and PO2 or even EtCO2 if measured, have to be provided in order to assess if the health of the horses could be compromised. In few words, when to start insufflation? Please add this important missing information.
L187-195. Table 2 and 3. Statistical differences are missing in these tables. Please add.
Response 13: Please provide further clarification as to your comment to enable us to address this point as requested. Thank you.
- It is important to add at least, an ANOVA of repeated measures to find any statistical differences over the time for the CV variables measured. Thank you.
Discussion and conclusion
As the question of investigation or the aim of the study Is not clear, the entire discussion section has to be rewritten.
Response 14: With further consideration of our revised aim of the investigation, please see the revised manuscript to determine if this is still your stance.
-Thank you.
